# Occurrence, Potential Sources, and Risk Assessment of Volatile Organic Compounds in the Han River Basin, South Korea

**DOI:** 10.3390/ijerph18073727

**Published:** 2021-04-02

**Authors:** Jong Kwon Im, Soon Ju Yu, Sujin Kim, Sang Hun Kim, Hye Ran Noh, Moon Kyung Kim

**Affiliations:** 1National Institute of Environmental Research, Han River Environment Research Center, 42, Dumulmeori-gil 68beon-gil, Yangseo-myeon, Yangpyeong-gun, Gyeonggi-do 12585, Korea; ysu1221@korea.kr (S.J.Y.); haemy@korea.kr (S.H.K.); anran1@korea.kr (H.R.N.); 2Department of Environmental Science, Center for Reservoir and Aquatic Systems Research, Baylor University, Waco, TX 76798, USA; Sujin_Kim@baylor.edu; 3Institute of Health and Environment, Seoul National University, 1 Gwanak-ro, Gwank-gu, Seoul 08826, Korea

**Keywords:** surface water, Seoul, ecological risks, WWTP, industrial complexes

## Abstract

Increasing public awareness about the aesthetics and safety of water sources has shifted researchers’ attention to the adverse effects of volatile organic compounds (VOCs) on humans and aquatic organisms. A total of 17 VOCs, including 10 volatile halogenated hydrocarbons and seven volatile non-halogenated hydrocarbons, were investigated at 36 sites of the Han River Basin, which is the largest and most important drinking water source for residents of the Seoul metropolitan area and Gyeonggi province in South Korea. The VOC concentrations ranged from below detection limits to 1.813 µg L^−1^. The most frequently detected VOC was 1,2-dichloropropane, with a detection frequency of 80.56%, as it is used as a soil fumigant, chemical intermediate, and industrial solvent. In terms of geographical trends, the sampling sites that were under the influence of sewage and industrial wastewater treatment plants were more polluted with VOCs than other areas. This observation was also supported by the results of the principal component analysis. In the present study, the detected concentrations of VOCs were much lower than that of the predicted no-effect concentrations, suggesting low ecological risk in the Han River. However, a lack of available ecotoxicity data and limited comparable studies warrants further studies on these compounds.

## 1. Introduction

As drinking water sources are frequently exposed to numerous pollutants generated by human and natural processes, it is very important to ensure safety. Moreover, as population and demand grow, investigations on the quality of drinking water sources and potential pollutants are essential [1,2]. Surface water sources contribute 78% of total water used in the Organization for Economic Cooperation and Development (OECD) countries [3]. In Korea, 87% of the water used by humans originates from surface water sources, such as streams, rivers, and lakes, whereas only ~13% of the water is obtained from groundwater sources [1]. Therefore, it is critical to investigate the concentrations of pollutants in surface waters and rivers. Furthermore, most of the urbanized and agricultural areas are located along rivers in Korea, and this causes the unavoidable reuse of effluents from the upstream regions by the people in the downstream regions. Globally, many studies have reported that sewage treatment plants are sources of various pollutants [4,5,6,7,8,9]. Therefore, Korea is focusing on the management of water quality in areas near surface water systems through various approaches [10].

Volatile organic compounds (VOCs) are carbon-containing organic chemicals with low water solubility and high vapor pressures under normal conditions, and hence, they can easily evaporate and enter the atmosphere [11,12,13]. VOCs are drained into the environment during their manufacturing, transportation, storage, use, and handling, and they can enter both surface water and groundwater from various non-point and point sources [14,15,16,17,18]. In addition, various VOCs are used in agricultural products, such as solvents for pesticides (i.e., 1,3-dichloropropene and xylene), herbicides (i.e., 1,4-dichlorobenzene and 1,2,4-trichlorobenzene), and fumigants (trichloroethane, naphthalene, 1,2-dichlorobenzene, 1,3-dichloropropene, 1,2-dichloropropane, and 1,2-dichloroethane) [19]. In addition, pentachloroethane is used as a degreaser, and diethylbenzenes as industrial solvents [19,20,21].

Reasons for high VOC concentrations in drinking water include oil leaks and spills from underground fuel/chemical storage tanks in urban areas and agricultural activities in rural areas [4,22,23]. In addition, VOCs can be released from plastic pipes in residential sewage distribution systems by leaching or from adhesives used in construction [22,23]. Furthermore, the disinfection process in drinking water treatment plants and the use of chemicals for specific treatments can both result in the nascence of specific VOC species [24]. Various studies have reported that chlorinated hydrocarbons are the most frequently detected VOCs in drinking water [21,25,26].

Although VOCs are mainly studied in atmospheric or indoor air quality research activities, concentration ranges have been reported in water quality monitoring activities such as in rivers [27,28], surface waters [29,30], and effluent [6,31,32]. Particularly, the ingestion of drinking water containing VOCs could cause damage to the immune, nervous, and reproductive systems, and lead to several types of cancers [20]. Because of the toxicity of VOCs to human health, the World Health Organization (WHO) and the United States Environmental Protection Agency (US EPA) have established maximum acceptable concentrations for some VOCs in drinking water based on its consumption over a lifetime [33,34].

The Han River Basin is the largest and most important source of drinking water to the residents of the Seoul metropolitan area and Gyeonggi province. However, limited investigations have been conducted on VOC concentrations in the watershed’s tributaries. Therefore, the objectives of this study were to report the levels of VOCs in the tributaries of the Han River Basin, compare the geographic trends of the detected VOCs at 32 sites, and assess the ecological risks of the VOCs using calculated risk quotients (RQs).

## 2. Materials and Methods

### 2.1. Chemicals and Reagents

A total of 17 compounds (listed in Appendix A) were investigated including cis-1,2-dichloroethene, trans-1,2-dichloroethene, cis-1,3-dichloropropene, trans-1,3-dichloropropene, hexachlorobutadiene, allyl chloride, epichlorohydrin, 1,2-dichloropropane, pentachloroethane, hexachloroethane, heptane, 2-methylhexane, nonane, 1-octene, 1,2-diethylbenzene, 1,3-diethylbenzene, and 1,4-diethylbenzene. Fluorobenzene, chlorbenzene-d5, and 1,4-dichlorobenzene-d4 were used as internal standards. All the chemicals were purchased from AccuStandard (New Haven, CT, USA).

Stock standard solutions and internal standards for each compound were prepared in methanol and stored at −20 °C. Fresh working solutions were used as spiking solutions and were generally used immediately after preparation. Ultrapure deionized water was obtained from a water filtration system (Purelab DV35, ELGA LabWater, Buckinghamshire, UK). Unless otherwise indicated, only analytical grade (or higher) chemicals were used.

### 2.2. Sampling Sites

This study focuses on the Han River Basin, which is the largest river basin (26,219 km^2^) in South Korea and accounts for approximately 27% of the country’s area. The river supplies drinking water to more than 26 million people [35]. It is 5417 km long and comprises two major branches, namely the Bukhan (BR; 10,652 km^2^) and Namhan (NR; 12,514 km^2^) rivers. The rivers gather right upstream of the Paldang Lake, a main source of water in the Seoul metropolitan area. The Hantan-Imjin rivers (HIR; 5943 km^2^) are located upstream of the Han River, and they join each other at the end of the Han River, which flows into the West Sea of South Korea. Each river has many small tributaries. The Han River Basin comprises forest, residential, business, and agricultural areas. Particularly, industrial complexes streams (ICS; 4.59 km^2^) are located near the Hantan-Imjin River and Asan Stream [9,36]. We obtained information on the Han River Basin from the national Water Management Information System [37].

River water samples were collected from 36 sites, which were chosen to represent diverse locations of the Han River Basin (Appendix A and Figure 1), including areas adjacent to ICS. In this study, 24 sewage treatment plants (STPs) and 7 industrial wastewater treatment plants (WWTPs) were identified as potential sources of VOCs, and details are shown in Appendix A. The samples were collected during three separate months of 2017, namely June, August, and October, and they were investigated for 17 volatile compounds.

### 2.3. Water Sampling and Analytical Procedures

Sampling campaigns were carried out on 23–27 June, 20–27 August, and 10–24 October 2017. For water sampling, brown glass bottles were used to prevent the occurrence of light-dependent reactions. Water sampling bottles were rinsed several times with ambient water before collecting samples. Surface water samples were collected manually in 40-mL and 1-L glass bottles without headspace at a depth of approximately 20 cm. The samples were maintained in iceboxes in the field before transfer to the laboratory, where they were stored in refrigerators at 4 °C until the analysis. The water quality data obtained at all sampling sites are presented in Appendix A.

The analysis of the 17 selected target compounds was performed using an AQUATeK 100 purge-and-trap (P&T) (TELEDYNE TERMAR, Mason, OH, UAS) and SCION SQ 456 Gas Chromatograph Mass Spectrometer (GC/MS; SCION Instrument, Scotland, UK) according to the US EPA Method 524.2 (US EPA, 1995). The characteristics of the analytical instruments are listed in Appendix A.

Quality assurance and quality control (QA/QC) was performed during the VOCs analysis, and a high precision level (R^2^ > 0.994) was achieved. Accuracy and relative standard deviations (RSD) ranged within 75.4−109.5% and 1.6−15.6%, respectively (Appendix A). The average method detection limits (MDL) for various chemicals was 0.0083 µg L^−1^ (Appendix A). These parameters confirmed the suitability of the laboratory analytical method. In addition, the inherent errors in the sampling process were addressed by performing the samplings in triplicate.

### 2.4. Risk Assessment

To evaluate the ecological risk of VOCs in river water, we calculated the RQs by dividing the highest measured environmental concentration (MEC) by the predicted no-effect concentration (PNEC) values. PNEC value was derived by dividing the toxicity data by the assessment factor (AF) while the AF was determined based on the technical guidance of the European Commission [38]. Ecotoxicity data on freshwater algae, crustaceans, and fish with the ecologically relevant endpoints (e.g., mortality, growth, reproduction, population, ecosystem) was obtained from the US Environmental Protection Agency (US EPA) ECOTOX database [39]. Data collection and risk assessment were performed only for the detected VOCs in the present study, which includes cis-1,2-dichloroethene, trans-1,2-dichloroethene, 1,2-dichloropropane, 1,3-diethylbenzene, 1,4-diethylbenzene, heptane, and hexachlorobutadiene. Among them, available toxicological information for 1,3-diethylbenzene, 1,4-diethylbenzene, and heptane suitable for the PNEC derivation were not found; therefore, subsequent analyses were not conducted.

### 2.5. Statistical Analysis

Statistical analyses were performed using SPSS 21 for Windows (SPSS, Inc., Chicago, IL, USA), and graphs were plotted using Sigmaplot 12.0 (Systat Inc., Point Richmond, CA, USA). The digital map of the Han River Basin was created in ArcGIS 9.2 (ESRI, Redlands, CA, USA). A one-way Analysis of Variance and two-samples paired t-test were performed to determine differences in VOCs levels. Principal Component Analysis (PCA) was used to describe similarities and differences between the studied areas, with respect to the major VOCs. *p*-values < 0.05 were considered to indicate significant difference.

## 3. Results and Discussion

### 3.1. Occurrence of VOCs

Tests were conducted to verify the presence of 17 VOCs in Han River Basin, but only seven VOCs i.e., cis-1,2-dichloroethene, trans-1,2-dichloroethene, hexachlorobutadiene, 1,2-dichloropropane, heptane, 1,3-diethylbenzene, 1,4-diethylbenzene were detected. Cis-1,3-dichloropropene, trans-1, 3-dichloropropene, allyl chloride, epichlorohydrin, pentachloroethane, hexachloroethane, 2-methylhexane, nonane, 1-octene, and 1,2-diethylbenzene were absent in the water collected in the present study. The concentrations of the detected target compounds are shown in Figure 2. The box plot depicts compounds of numerical data through their six-number summaries (minimum observation, lower quartile, medium, upper quartile, and maximum observation). The six parameters are only given for values higher than their MDLs.

Table 1 summarizes the analysis results of the water samples. The concentrations of the seven detected VOCs varied from 0.0006 to 1.8131 µg L^−1^. Among them, the frequency of 1,2-dichloropropane was the highest, followed by those of cis-1,2-dichloroethene, trans-1,2-dichloroethene, and 1,4-diethylbenzene. In contrast, hexachlorobutadiene and 1,3-diethylbenzene existed specifically in just one sample.

The concentrations of cis 1,2-dichloroethene obtained in the present study (0.0022−0.4580 µg L^−1^) were lower than those obtained in a previous study (0.0155−2.6800 µg L^−1^) [1]. According to Cho et al. [1], the concentration of hexachlorobutadiene in the Han River ranged from 0.0290 to 0.0670 µg L^−1^, which is 100 times higher than the values in our study [1]. In addition, the reported levels of VOCs in other countries such as Italy (N.D−0.44 µg L^−1^ for 1,2-dichloropropane) [40], Russia (<0.10−0.46 µg L^−1^ for cis 1,2-dichloroethene) [41], China (0.01−1.23 µg L^−1^ for hexachlorobutadiene) [2], Kuwait (N.D−1.71 µg L^−1^ for 1,2-dichloropropane) [42], Taiwan (0.16−0.57 µg L^−1^ for hexachlorobutadiene) [32], and Belgium (1.60−6.60 µg L^−1^ for trans 1,2-dichloroethene and 0.87−5.60 µg L^−1^ for 1,2-dichloropropane) [43] vary significantly as compared to those in the present study. Among the other countries, in particular, the maximum concentration of 1,2-dichloropropane in China was approximately 1 µg L^−1^, which is two times higher than that in our study [2]. Other countries mostly reported non-detection or LOD concentrations [41,43], which were similar or lower than those reported in the present study [40,42]. Additionally, the concentrations of hexachlorobutadiene in waters were lower in the present study than in the five major river basins of China [2], Italy [40], and Taiwan [32]. In contrast, the concentrations of 1,2-dichloroethene in Chinese river waters ranged from < 0.06 to 0.76 µg L^−1^ (average of 0.07 µg L^−1^), which are only slightly higher or similar to those of the present study [21]. However, it was not detected in rivers and lakes in Greece, and had a maximum concentration of 5.3 µg L^−1^ in STPs [31]. As such, various VOC detection concentrations in each country are likely to be caused by point or non-point pollution sources near the sampling sites, and seem to be closely related to the usage of VOC compounds in each country. In other words, the variation can be attributed to the site-specific distribution of VOC compounds in each country.

There are no standard values for 1,2-dichloropropane, cis-, and trans-1,2-dichloroethene in the National Drinking Water Quality Standards or National Environmental Quality Standards for Surface Water of South Korea. However, the maximum concentrations of 1,2-dichloropropane in drinking water set by other countries (i.e., Japan [44] and the United States [34] and the World Health Organization (WHO) range from 0.04 to 0.06 mg L^−1^ [33]. In the case of cis-, and trans-1,2-dichloroethene, the maximum concentrations in drinking water established by other countries (i.e., Japan and the United States) range from 0.04 to 0.07 L^−1^ [34] (See Appendix A).

In order to evaluate the differences among the three sampling periods, we used analysis of variance (Kruskal-Wallis H test). The result showed that only 1,2-dichloropropane was significantly correlated with the temporal distribution in the Han River water system (*p* = 0.003). Maximum concentration of 1,2-dichloropropane was observed in August (0.1001 ± 0.3012 μg L^−1^).

For comparison, the 17 measured VOCs were classified into two groups: (1) volatile halogenated hydrocarbons, including cis-1,2-dichloroethene, trans-1,2-dichloroethene, cis-1,3-dichloropropene, trans-1,3-dichloropropene, hexachlorobutadiene, allyl chloride, epichlorohydrin, 1,2-dichloropropane, pentachloroethane, and hexachloroethane; and (2) volatile non-halogenated hydrocarbons, including heptane, 2-methylhexane, nonane, 1-octene, 1,2-diethylbenzene, 1,3-diethylbenzene, and 1,4-diethylbenzene. Volatile halogenated hydrocarbons were the most frequently detected VOC group in this study. Many surveys have shown that chlorination byproducts and chlorinated solvents are VOCs frequently detected in water bodies [42,45,46,47]. Overall, the concentrations of VOCs in the Han River water system were estimated to be similar or slightly higher than those of other countries [6,27,31,32,40,41,42,43]. However, continuous monitoring is required to systematically manage VOCs and establish drinking water standards.

### 3.2. Geographic Trends

The concentrations of target VOCs at different sampling sites are shown in Figure 2. The total VOCs concentrations at sampling sites BR1-5, NR1-7, HR1-8, HIR1-4, AS1, and ICS1-11 were 0.0360 (average 0.0045), 0.2282 (average 0.0095), 1.0053 (average 0.0197), 3.5778 (average 0.1404), 0.1497 (average 0.0250), and 3.3997 µg L^−1^ (average 0.0485 µg L^−1^), respectively.

As shown in Figure 3, 1,2-dichloropropane was consistently found in all sampling sites. However, it was not reported in the domestic pollutant release and transfer register system [48], possible point sources of pollution are not known. Nevertheless, this substance is known to be used as a soil fumigant, chemical intermediate, and industrial solvent [49]. For this reason, 1,2-dichloropropane was frequently detected in all sampling sites.

Among the HIR areas, the total VOC concentrations in HIR-1 were the highest at 3.0153 μg L^−1^, followed by HIR-2 (0.3842 μg L^−1^), HIR-3 (0.1473 μg L^−1^), and HIR-4 (0.0309 μg L^−1^). HIR-1 is located 12 km downstream of a small WWTP (1 × 10^3^ m^3^ d^−1^ capacity), and it is close to tourist destinations, including camping facilities, family parks, sports facilities, and a prehistoric site. Therefore, the VOC sources at the location may be more complex. However, the high 1,2-dichloropropane concentrations could be attributed to surface runoff of fumigants used to manage recreational facilities around HIR-1.

The total VOC concentrations in the eight-site HR area ranged from 0.0182 to 0.6122 μg L^−1^, depending on the sampling location. Unlike in other HR locations, the concentrations of VOCs in HR-5 and HR-7 were relatively high because they were located downstream of large-scale WWTPs. Such sites receive treated wastewater from WWTP-20, -21 (128 × 10^3^ and 236 × 10^3^ m^3^ d^−1^ capacity) and WWTP-23 (123 × 10^3^ m^3^ d^−1^ capacity), and the total concentrations in two sites were 0.6122 and 0.1689 μg L^−1^, respectively. The results are similar to the findings of other studies investigating VOCs in WWTPs, and suggest that WWTPs mainly contribute to VOC pollution in surface waters [4,6,11,31].

Within the NR area, the VOC concentrations were relatively high in NR-5, which is one of the major tributaries located at the lower part of the Bokha stream. NR-5 has a high level of water pollution caused by scattered non-point sources such as livestock complexes and farmlands [50]. A WWTP (42 × 10^3^ m^3^ day^−1^ of capacity) was located upstream of the NR-5 site. Therefore, the VOC concentrations at the site are the result of a complex mix of pollutants. Other locations, such as NR-3, -4, -6, and -7, displayed similar concentrations and distributions of VOCs, whereas NR-1 and -2 had the lowest concentrations. The results were considered a product of water quality regulations by local governments, as there were no upstream WWTPs, and the areas around NR-1 (706,000 m^2^) and NR-2 (2,317,000 m^2^) have been designated water source protection zones since 1989 and 1981, respectively [35].

### 3.3. Potential Sources

In South Korea, the HIR and ICS areas were more polluted with VOCs than the other areas of the Han River Basin (Figure 4a). As shown in Appendix A, the highest VOC concentrations were recorded in the HIR area, with a 9.8% average detection frequency, and relatively high concentrations were detected in the ICS area, with an 11.59% average detection frequency. The results could be attributed to the activity of the industrial complex near the sampling site. Moreover, the average total VOC concentrations were mainly high at the sampling sites downstream of the STPs and industrial WWTPs. Therefore, the VOCs in the Han River Basin are potentially associated with STPs and industrial WWTPs.

PCA which is often used in various media such as water [30,51], atmosphere [13], soil [52] and sediment [53] to track pollutants, was adopted for the identification of the sources of VOCs in water in each area. The PCA results showed that principal component (PC) 1, PC 2, and PC 3 contributed 44.63%, 35.00%, and 16.70% of the total variance, respectively (Figure 4b). The BR and AS watersheds had a strong positive loading on PC1, indicating that the respective sampling stations had similar pollution profiles. The sites around the areas were among the cleanest, with respect to VOCs, which presented an average detection frequency of 3.76%, with 1,2-dichloropropane as the primary compound detected. The result seems reasonable due to the absence of industrial complexes in the areas. The HIR and ICS watersheds had strong positive loadings on PC2, and PC2 had a strong correlation with cis-1,2-dichloroethene and 1,2-dichloropropane, as the major compounds detected.

As shown in Figure 5, the levels of VOCs in the rural, urban, and industrial areas were significantly different (*p* < 0.05). The respective average total concentrations of VOCs in the areas were 0.0117 ± 0.0190, 0.1031 ± 0.2882, and 0.0494 ± 0.0560 μg L^−1^. STP 13 also treats industrial wastewater, which could potentially affect the VOC concentrations in sites ICS-6 and -7. The results indicate that the main sources of VOCs in surface water are anthropogenic point sources.

Urban and industrial environments provide potential complex sources of VOCs to surface water. Many industrial processes produce raw wastewater containing VOCs, which is sent to wastewater treatment plants for removal. However, in the wastewater treatment process, some VOCs are discharged without being completely removed [5,7,8]. Road runoff and urban stormwater potentially introduce VOC loads to surface waters [16,17,54]. In particular, urban storm water is known to be affected by contaminated land, which generally presents higher VOC concentrations than the atmosphere [12,17]. In addition, seepage of VOC-contaminated groundwater into rivers can be a source of surface water pollution over the long term [14,15,16,18,55].

### 3.4. Risk Assessment of VOCs

Derivations of the PNEC and RQ values are presented in Table 2 and Appendix A, respectively. Our risk assessment results show that those calculated here are much lower than one (ranged from 0.0000489 to 0.0151), implying RQs for cis- and trans-1,2-dichloroethene, hexachlorobutadiene, and 1,2-dichloropropane with negligible risk of the VOCs in the Han River. However, the results should be taken with caution because of high uncertainty in the determination of the PNEC values, which resulted from a lack of available ecotoxicity data for such compounds. For example, as shown in Appendix A, AFs for both cis- and trans-1,2-dichloroethene were determined to be 1000 due to the absence of long-term data under our search criteria. More ecotoxicity studies based on standardized testing protocols need to be conducted. Similarly, for other detected VOCs, such as 1,3-diethylbenzene, 1,4-diethylbenzene, and heptane, ecological risk was not estimated because of lack of ecotoxicity data suitable for the assessment, which warrants further investigations, particularly following long-term exposure to such VOCs in the environment.

## 4. Conclusions

Seventeen VOCs were observed at 36 sites in the largest and most important river tributaries used as a drinking water source for the people of Gyeonggi Province and Seoul metropolitan area in South Korea. Seven VOCs were mainly detected, with 1,2-dichloropropane and cis-1,2-dichlorothene being the most frequently detected, at 80.56% and 58.78%, respectively. Except for 1,2-dichloropropane, 1,2-dichloroethene, and diethylbenzene, VOC compounds have rarely been observed in the Han River Basin, but their pattern of concentrations must be tracked by additional monitoring. Geographically, the HIR and ICS areas were more contaminated with VOCs than other areas in the Han River Basin, perhaps due to the activity of adjacent industrial complexes. As a consequence of the PC study, possible sources of VOCs appear to be correlated with STPs and industrial WWTPs. According to the finding of the study, the quality of surface water in urban areas is directly linked to the discharge of sewage and industrial wastewater effluents. The detected VOC concentration was estimated to pose a very low ecological risk in terms of ecological risk assessment, but further research into the possibility of ecotoxicity after long-term exposure is required.

## Figures and Tables

**Figure 1 ijerph-18-03727-f001:**
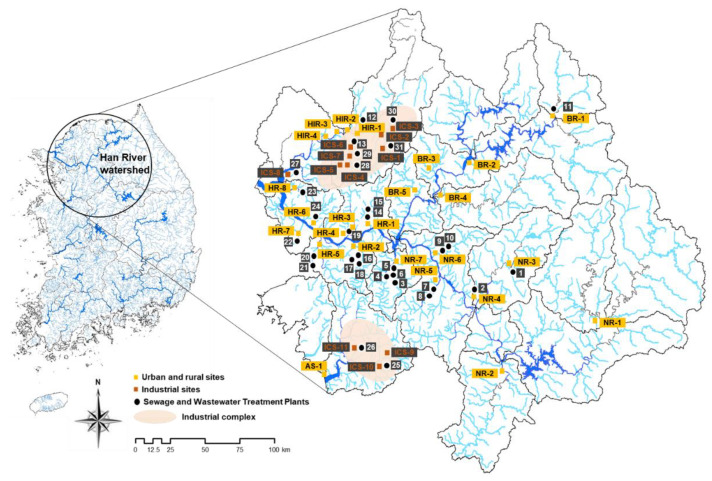
Location of sampling sites in the study area.

**Figure 2 ijerph-18-03727-f002:**
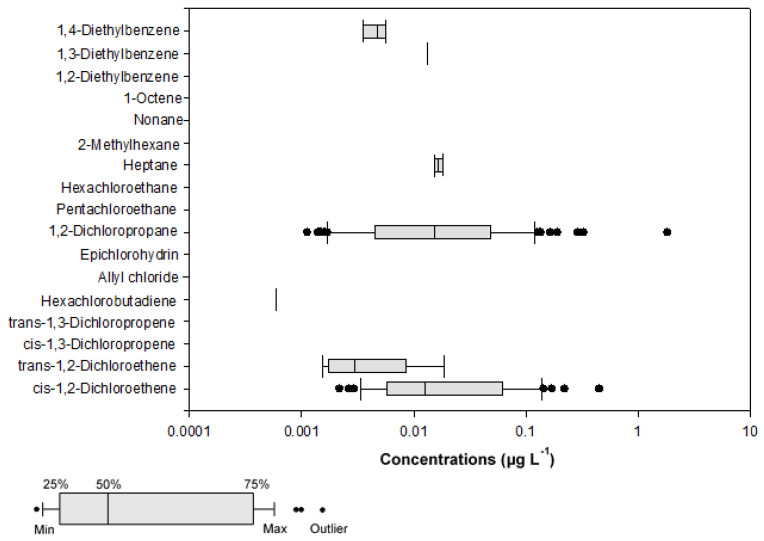
Box plot of volatile organic compounds (VOC) concentrations in the Han River Basin.

**Figure 3 ijerph-18-03727-f003:**
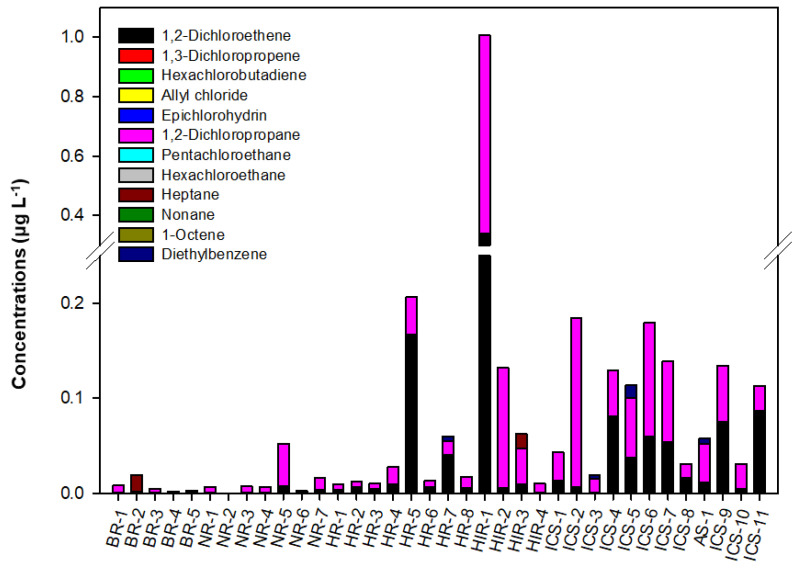
Average concentration of VOCs at different sampling sites. 1,2-dichloroethene includes cis- and trans-1,2-dichloroethene; 1,3-dichloropropene includes cis- and trans-1,3-dichloropropene; heptane includes 2-methylhexane isomer; and diethylbenzene includes 1,2-, 1,3-, and 1,4-diethylbenzene isomers.

**Figure 4 ijerph-18-03727-f004:**
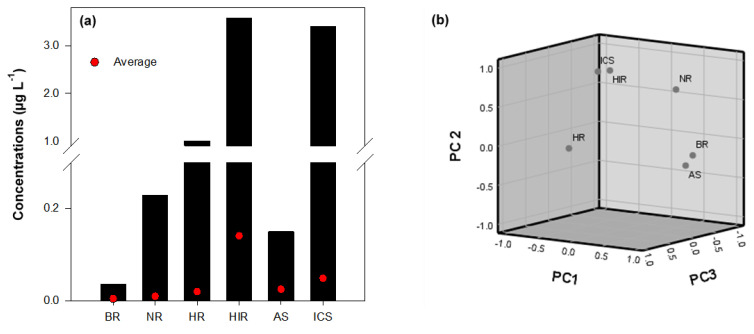
(**a**) Cumulative concentrations of VOCs at different sampling sites, and (**b**) loading plot of principal component analysis (PCA) analysis. BR: Bukhan River, NR: Namhan River, HR: Han River, HIR: Hantan-Imjin River, AS: Anseong Stream, ICS: Industrial complex Stream.

**Figure 5 ijerph-18-03727-f005:**
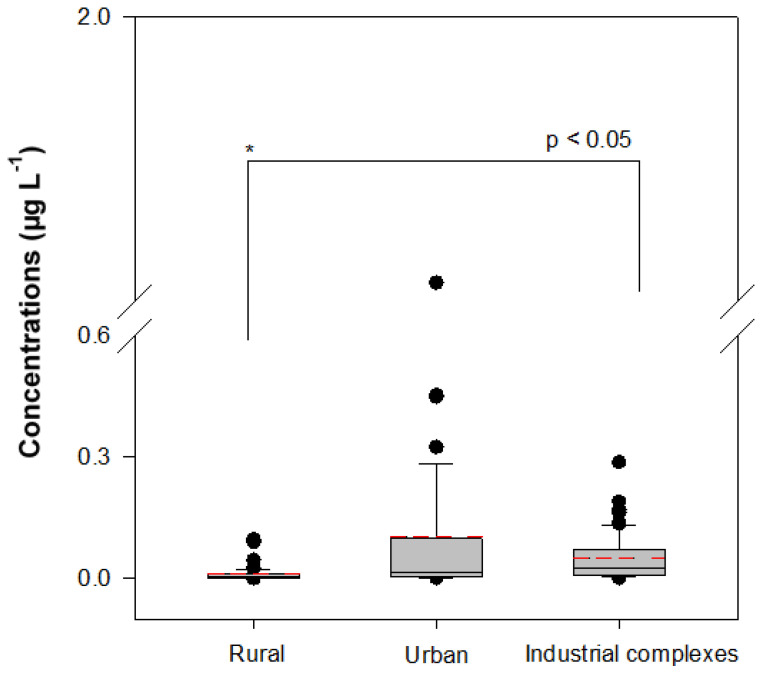
Distributions of VOCs concentrations at potential pollution sources in rural area (sites BR-1 to -5, NR-1 to -7, HIR-1 to -4, and AS-1), urban area (sites HR-1 to -8), industrial complex areas (sites ICS-1 to -11). In the box plot, the boxes indicate 25–75% quartile ranges, central line indicates medians, whiskers indicate data ranges, and red dotted lines indicate average concentrations.

**Table 1 ijerph-18-03727-t001:** Measured VOC concentrations in the sampling period.

Compound	Min (µg L^−1^)	Max (µg L^−1^)	Average ± SD (µg L^−1^)	DF (%)
cis-1,2-Dichloroethene	0.0022	0.4528	0.0523 ± 0.0909	52.78
trans-1,2-Dichloroethene	0.0016	0.0187	0.0055 ± 0.0066	5.56
cis-1,3-Dichloropropene	ND	ND	ND	ND
trans-1,3-Dichloropropene	ND	ND	ND	ND
Hexachlorobutadiene	0.0006	0.0006	0.0006	0.93
Allyl chloride	ND	ND	ND	ND
Epichlorohydrin	ND	ND	ND	ND
1,2-Dichloropropane	0.0011	1.8131	0.0595 ± 0.1989	80.56
Pentachloroethane	ND	ND	ND	ND
Hexachloroethane	ND	ND	ND	ND
Heptane	0.0153	0.0180	0.0167 ± 0.0019	1.85
2-Methylhexane	ND	ND	ND	ND
1-Octene	ND	ND	ND	ND
Nonane	ND	ND	ND	ND
1,2-Diethylbenzene	ND	ND	ND	ND
1,3-Diethylbenzene	0.0133	0.0133	0.0133	0.93
1,4-Diethylbenzene	0.0036	0.0056	0.0046 ± 0.0010	2.78
**Total VOCs**			**0.0525 ± 0.1581**	

ND: Not Detected; DF: Detection Frequency. Total VOCs: Sum of individual VOCs (such as ΣVOC).

**Table 2 ijerph-18-03727-t002:** Risk quotients (RQs) from predicted no-effect concentration (PNEC) and measured environmental concentration (MEC) for VOCs at Han River Basin.

Compound	Lowest PNEC (μg L^−1^)	Toxicity Endpoint	Max. MEC (μg L^−1^)	RQs	References
cis-1,2-Dichloroethene	59.7	*Pseudokirchneriella subcapitata* Population, EC_50_	0.4528	0.0076	[56]
trans-1,2-Dichloroethene	36.4	*Pseudokirchneriella subcapitata* Population, EC_50_	0.0187	0.0005	[56]
Hexachlorobutadiene	0.130	*Pimephales promelas* Mortality, NOEC	0.0006	0.0046	[57]
1,2-Dichloropropane	120	*Pimephales promelas* Growth, NOEC	1.8131	0.0151	[58]

## Data Availability

Not applicable.

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
