# Peer review of "Occurrence, Potential Sources, and Risk Assessment of Volatile Organic Compounds in the Han River Basin, South Korea"

_ijerph, 2021, doi:10.3390/ijerph18073727_

Round 1
Reviewer 1 Report
The manuscript deals with: "Occurrence, Potential Sources, and Risk Assessment of Volatile Organic Compounds in the Han River Basin, South Korea". This study report the levels of VOCs in the tributaries of the Han River Basin, and compare the geographic trends of the detected VOCs at 32 sites. The author also assess the ecological risks of the VOCs using calculated risk quotients (RQs). Overall, the manuscript is well written and on a topic of relative importance that falls within the scope of the journal. There are a few minor issues around the text that need fixing or better clarification.
1. Introduction sections are well-presented.
2. Line 114 Please include information on how the water samples are taken?
3. Line 176-178 It would be better to present the ranges of VOCs concentrations.
4. Line 178-187 Can you explain briefly these variations in VOCs levels.
5. Line 280-281 Delete the "complex".
6. Line 318 The conclusion should not the repeated of the main results, it should be more concise and summary.
Author Response
Dear Reviewer 1
Specific comments
The manuscript deals with: "Occurrence, Potential Sources, and Risk Assessment of Volatile Organic Compounds in the Han River Basin, South Korea". This study report the levels of VOCs in the tributaries of the Han River Basin, and compare the geographic trends of the detected VOCs at 32 sites. The author also assess the ecological risks of the VOCs using calculated risk quotients (RQs). Overall, the manuscript is well written and on a topic of relative importance that falls within the scope of the journal. There are a few minor issues around the text that need fixing or better clarification.
Q1. Introduction sections are well-presented.
Line 114 Please include information on how the water samples are taken?
|
Answer: Thank you for the helpful comment. We added more information on water sampling method for help to the readers in the revised text as follows;
From: “For sampling, brown glass bottles were used to avoid the occurrence of light-dependent reactions. The samples were collected in 40 mL and 1 L glass bottles without headspace.”
To: “For water sampling, brown glass bottles were used to prevent the occurrence of light-dependent reactions. Water sampling bottles were rinsed several times with ambient water before collecting samples. Surface water samples were collected manually in 40 mL and 1 L glass bottles without headspace at a depth of approximately 20 cm.” (Line 118-121) |
Q2. Line 176-178 It would be better to present the ranges of VOCs concentrations.
|
Answer: Thank you for the comment. As the reviewer suggested, we added the reported VOCs concentrations in other countries in the revised text as follows;
From: “In addition, the reported levels of VOCs in other countries such as Italy [40], Russia [41], China [2], Kuwait [42], Taiwan [32] and Belgium [43] vary significantly compared to those in the present study.”
To: “In addition, the reported levels of VOCs in other countries such as Italy (N.D−0.44 µg L−1 for 1,2-dichloropropane) [40], Russia (<0.10−0.46 µg L−1 for cis 1,2-dichloroethene) [41], China (0.01−1.23 µg L−1 for hexachlorobutadiene) [2], Kuwait (N.D−1.71 µg L−1 for 1,2-dichloropropane) [42], Taiwan (0.16−0.57 µg L−1 for hexachlorobutadiene) [32] and Belgium (1.60−6.60 µg L−1 for trans 1,2-dichloroethene and 0.87−5.60 µg L−1 for 1,2-dichloropropane) [43] vary significantly as compared to those in the present study.” (Line 186-192) |
Q3. Line 178-187 Can you explain briefly these variations in VOCs levels.
|
Answer: Thank you for the comment. As the reviewer suggested, we explained briefly these variations in VOCs levels in each country as follows; “As such, various VOC detection concentrations in each country are likely to be caused by point or non-point pollution sources near the sampling sites, and seem to be closely related to the usage of VOC compounds in each country. In other words, the variation can be attributed to the site-specific distribution of VOC compounds in each country.” (Line 201-205)
|
Q4. Line 280-281 Delete the "complex".
|
Answer: Thank you for having us clarify this. As the reviewer suggested, we deleted the word in the revised text as follows;
From: “…., the levels of VOCs in the rural, urban, and industrial complex areas were….” To: “...., the levels of VOCs in the rural, urban, and industrial areas were…” (Line 301) |
Q5. Line 318 The conclusion should not the repeated of the main results, it should be more concise and summary.
|
Answer: Thank you for the comment. As the reviewer suggested, we rewrote the conclusion in the revised text as follows;
From: “In the present study, we investigated 17 target VOCs in the largest and most important drinking water sources to the residents of the Seoul metropolitan area and Gyeonggi province in South Korea. The obtained VOC concentrations ranged from below laboratory MDL to 1.8131 µg L−1 (1,2-dichloropropane). The two most frequently detected VOCs were 1,2-dichloropropane and cis-1,2-dichloroethene, with respective detection frequencies of 80.56% and 58.78%. Except for 1,2-dichloropropane, 1,2-dichloroethene, and diethylbenzenes, most VOCs were hardly detected in the tributaries of the Han River Basin. The presence of 1,2-dichloropropane was consistently observed in all sampling sites, as it is extensively used as a soil fumigant, chemical intermediate, and industrial solvent. In terms of geographical trends, the HIR and ICS areas were more polluted with VOCs than other areas in the Han River Basin, which may be attributed to the activity of nearby commercial complexes. Therefore, potential complex sources of VOCs seem to be associated with STPs and industrial WWTPs as a result of PC analysis. The results of this study suggest that urban surface water quality in urban areas is closely related with sewage and industrial wastewater effluent management and discharge. Although the ecological risks of studied VOCs in the Han River were estimated to be negligible in the present study, ecotoxicity potential, particularly following long-term exposure, needs to be examined in the future.”
To: “Seventeen VOCs were observed at 36 sites in the largest and most important river tributaries used as a drinking water source for the people of Gyeonggi Province and Seoul metropolitan area in South Korea. Seven VOCs were mainly detected, with 1,2-dichloropropane and cis-1,2-dichlorothene being the most frequently detected, at 80.56 % and 58.78 % respectively. Except for 1,2-dichloropropane, 1,2-dichloroethene, and diethylbenzene, VOC compounds have rarely been observed in the Han River Basin, but their pattern of concentrations must be tracked by additional monitoring. Geographically, the HIR and ICS areas were more contaminated with VOCs than other areas in the Han River Basin, perhaps due to the activity of adjacent industrial complexes. As a consequence of the PC study, possible sources of VOCs appear to be correlated with STPs and industrial WWTPs. According to the finding of the study, the quality of surface water in urban areas is directly linked to discharge of sewage and industrial wastewater effluent. The detected VOC concentration was estimated to pose a very low ecological risk in terms of ecological risk assessment, but further research into the possibility of ecotoxicity after long-term exposure is required.” (Line 340-354) |
We have addressed all the reviewer’s comments and corrections in the text and would like to thank them for his/her assistance.
Best regards
Jongkwon IM

Reviewer 2 Report
The authors reported in this study the concentrations of VOCs in a watershed of Korea.
The research is well-designed, although the content is not very new. The reviewer suggests minor revisions before the publication of this paper.
1) It is difficult for the readers to understand that Figure 2 is drawn as a cumulative bar graph. The space between black and pink bars at HIR-1 position make the readers confuse. Reformat figure 2 to avoid misunderstandings.
2) Specify units (ng/L or micro-g/L) of concentration in Table 1.
3) Check units in Table 2. The reviewer thinks that ng/L is a mistake and the correct unit would be micro-g/L.
4) The legend for most right column in Table S6 is missing. In addition, check the QI and CI values in this table, because the deviations from integer numbers for QI and CI seem to be larger than those for usual SIM in GC/MS operation.
5) Sampling details such as sampling locations (Figure S1) have to be explained in the main text. In addition, the dates of sampling are not clearly written. Judging from the temperatures of the samples (Table S4), seasonal variations must be discussed because the concentrations of VOCs in water must partially be determined by equilibrium between air and water.
Author Response
Dear Reviewer 2
Specific comments
The authors reported in this study the concentrations of VOCs in a watershed of Korea.
The research is well-designed, although the content is not very new. The reviewer suggests minor revisions before the publication of this paper.
Q1. It is difficult for the readers to understand that Figure 2 is drawn as a cumulative bar graph. The space between black and pink bars at HIR-1 position make the readers confuse. Reformat figure 2 to avoid misunderstandings.
|
Answer: Thank you for having us clarifying this. As the reviewer suggested, we reformatted Fig.3 (it was Fig.2 in original version) in the revised text as follows;
Figure 3. Average concentration of VOCs at different sampling sites. 1,2-dichloroethene includes cis- and trans-1,2-dichloroethene; 1,3-dichloropropene include cis- and trans-1,3-dichloropropene; heptane includes 2-methylhexane isomer; and diethylbenzene includes 1,2-, 1,3-, and 1,4-diethylbenzene isomers. (Line 244-248) |
Q2. Specify units (ng/L or micro-g/L) of concentration in Table 1.
|
Answer: Thank you for the comment. As the reviewer suggested, we wrote the units (µg L−1) in Table 1 in revised text as follows;
Table 1. Measured VOCs concentrations in the sampling period.
|
Q3. Check units in Table 2. The reviewer thinks that ng/L is a mistake and the correct unit would be micro-g/L.
|
Answer: Thank you for your kind suggestion. We made a mistake in filling out units in Table 2. That has been modified in the revised text as follows;
Table 2. Risk quotients (RQs) from predicted no-effect concentration (PNEC) and measured environmental concentration (MEC) for VOCs at Han River Basin.
|
Q4. The legend for most right column in Table S6 is missing. In addition, check the QI and CI values in this table, because the deviations from integer numbers for QI and CI seem to be larger than those for usual SIM in GC/MS operation.
|
Answer: Thank you for the comment. As the reviewer suggested, we checked the QI and CI values in Table S6. The legend in the most right column represents the confirmation ion. We don’t know why the deviations from integer numbers for QI and CI were large; however, the QI and CI values are correct. |
Q5. Sampling details such as sampling locations (Figure S1) have to be explained in the main text. In addition, the dates of sampling are not clearly written. Judging from the temperatures of the samples (Table S4), seasonal variations must be discussed because the concentrations of VOCs in water must partially be determined by equilibrium between air and water.
|
Answer: Thank you for helpful comment. As the reviewer suggested, we have revised as follows; [Sampling locations] “Figure S1 is moved to main text and the explanation of the sampling site has already been written.” (Line 115) [Date of sampling] “Sampling campaigns were carried out on June 23-27th, August 20-27th, and October 10-24th, 2017.” (Line 117-118) [Seasonal variations] “In order to evaluate the differences among the three sampling periods, we used analysis of variance (Kruskal-Wallis H test). The result showed that only 1,2-dichloropropane was significantly correlated with the temporal distribution in the Han River water system (p = 0.003). Maximum concentration of 1,2-dichloropropane was observed in August (0.1001 ± 0.3012 μg L−1).” (Line 214-218) |
We have addressed all the reviewer’s comments and corrections in the text and would like to thank them for his/her assistance.
Best regards
Jongkwon Im
